# Pseudo-pac site sequences used by phage P22 in generalized transduction of *Salmonella*

**Jessie L. Maier** [1]*, **Craig Gin**[2], **Benjamin Callahan**[2], **Emma K. Sheriff**[3], **Breck A. Duerkop**[3], **Manuel Kleiner** [1]*

**1** Department of Plant and Microbial Biology, North Carolina State University, Raleigh, North Carolina, United States of America, **2** Department of Population Health and Pathobiology, North Carolina State University, Raleigh, North Carolina, United States of America, **3** Department of Immunology and Microbiology, University of Colorado - Anschutz Medical Campus, School of Medicine, Aurora, Colorado, United States of America

* jlmaier@ncsu.edu (JLM); manuel_kleiner@ncsu.edu (MK)

**Data Availability Statement:** P22 sequencing reads used for LT2 read mapping were generated in Kleiner et al. (2020) https://doi.org/10.1186/s40168-020-00935-5 and are available at ENA

## Abstract

*Salmonella enterica* Serovar Typhimurium (*Salmonella*) and its bacteriophage P22 are a model system for the study of horizontal gene transfer by generalized transduction. Typically, the P22 DNA packaging machinery initiates packaging when a short sequence of DNA, known as the pac site, is recognized on the P22 genome. However, sequences similar to the pac site in the host genome, called pseudo-pac sites, lead to erroneous packaging and subsequent generalized transduction of *Salmonella* DNA. While the general genomic locations of the *Salmonella* pseudo-pac sites are known, the sequences themselves have not been determined. We used visualization of P22 sequencing reads mapped to host *Salmonella* genomes to define regions of generalized transduction initiation and the likely locations of pseudo-pac sites. We searched each genome region for the sequence with the highest similarity to the P22 pac site and aligned the resulting sequences. We built a regular expression (sequence match pattern) from the alignment and used it to search the genomes of two P22-susceptible *Salmonella* strains—LT2 and 14028S—for sequence matches. The final regular expression successfully identified pseudo-pac sites in both LT2 and 14028S that correspond with generalized transduction initiation sites in mapped read coverages. The pseudo-pac site sequences identified in this study can be used to predict locations of generalized transduction in other P22-susceptible hosts or to initiate generalized transduction at specific locations in P22-susceptible hosts with genetic engineering. Furthermore, the bioinformatics approach used to identify the *Salmonella* pseudo-pac sites in this study could be applied to other phage—host systems.

## Author summary

Bacteriophage P22 has been a genetic tool and a key model for the study of generalized transduction in *Salmonella* since the 1950s, yet certain components of the generalized transduction molecular mechanism remain unknown. Specifically, the locations and sequences of pseudo-pac sites, hypothesized to facilitate packaging of *Salmonella* DNA by P22, have not been determined to date. In this study, we identified the specific locations

(Study: PRJEB6941, Sample: SAMEA2690949).
P22 sequencing reads used for 14028S read
mapping are available at ENA (Study: PRJEB72417,
Sample: SAMEA115180785). The reference
genomes used for LT2 and 14028S read mapping
are from NCBI RefSeq NC_003197.2 and
NC_016856.1, respectively.

**Funding:** This work was supported by funding
from the NC State University Data Science
Academy and by the National Institute of General
Medical Sciences and the National Institute of
Allergy and Infectious Diseases of the National
Institutes of Health under Award Number
R35GM138362 (MK) and R01AI141479 (BAD).
The content is solely the responsibility of the
authors and does not necessarily represent the
official views of the National Institutes of Health or
NCSU Data Science Academy. The funders had no
role in study design, data collection and analysis,
decision to publish, or preparation of the
manuscript. NCSU DSA: https://
datascienceacademy.ncsu.edu/seed-grants/
NIGMS: https://www.nigms.nih.gov/ NIAID: https://
www.niaid.nih.gov/ NIH: https://www.nih.gov/.

**Competing interests:** The authors have declared
that no competing interests exist.

and sequences of the pseudo-pac sites frequently recognized by P22 in *Salmonella* genomes. The identification of highly efficient pseudo-pac sites in *Salmonella* helps us understand the sequence specificity necessary for P22 pac site recognition and paves the way for more targeted use of generalized transduction with P22.

## Introduction

Transduction, the transfer of DNA between bacterial cells by bacteriophages, can lead to horizontal gene transfer of entire operons of genetic material and can cause dramatic changes in bacterial phenotypes [1–4]. Generalized transduction, one of several potential modes of transduction, was first discovered in 1952 in the bacteriophage P22 and *Salmonella enterica* Serovar Typhimurium LT2 (LT2), thus making these a model system for generalized transduction [5,6]. Generalized transduction was initially thought to be a random transfer of host DNA, but when the frequencies of transduced LT2 gene markers were quantified, it became clear that transduction frequencies differed widely across the genome [7–9]. Similar transduction locations and frequencies were seen in 2 recent studies that used mapped P22 DNA sequencing reads to visualize transduction patterns in LT2. These studies demonstrate that the P22-facilitated generalized transduction of LT2 is nonrandom and consistent between methods and experiments [10,11]. The observed pattern consists of sharp increases in read coverage followed by sloping decreases of coverage across several regions of the LT2 genome (Fig 1A).

P22 uses a headful packaging mechanism in which a short sequence of DNA, known as the pac site, is recognized prior to packaging initiation. After initiation, the P22 DNA packaging machinery packages several capsids in series using the same concatemer of DNA on which the pac site is located [14–16]. Generalized transduction by P22 occurs when its small terminase, responsible for pac site recognition [17,18], recognizes a sequence in the host genome that is similar to the phage's pac site (i.e., pseudo-pac site), leading to initiation of packaging on the bacterial chromosome [9,18–20]. The locations of pseudo-pac sites along a bacterial chromosome lead to the nonrandom generalized transduction patterns observed between P22 and LT2. Despite P22's pac site sequence having been previously described by Wu and colleagues [13], the exact pseudo-pac site sequences and locations remained unknown.

## Results

### Identifying pseudo-pac site candidate sequences

We used sequencing reads from ultra-purified P22 propagated on LT2 and mapped them back to the LT2 genome to identify the locations where pseudo-pac site facilitated packaging of the LT2 genome occurred. The regions where packaging is initiated are characterized by a sudden, sharp increase in read coverage (Fig 1A). We visually identified 8 sites that matched this profile and extracted 120 base pair (bp) regions of the LT2 genome surrounding these sites (Fig 1A). We chose 120 bp because the P22 packaging machinery makes its packaging initiation cuts in a 120 bp region surrounding its pac site [17]. We searched each of the eight 120 bp regions for the sequence that best matched the 12 bp P22 consensus pac site sequence—5′ AAGATTTAT CTG 3′—identified in Casjens and colleagues [12] and further characterized by Wu and colleagues [13] using P22 mutants. For generalized transduction events whose read coverage patterns sloped left to right across the LT2 genome (sites 3, 4, 5, 7, and 8), we searched the forward strand of the genome and for events that sloped right to left (sites 1, 2, and 6), we searched the reverse strand. The 8 sequences that best matched the P22 pac site in each of the

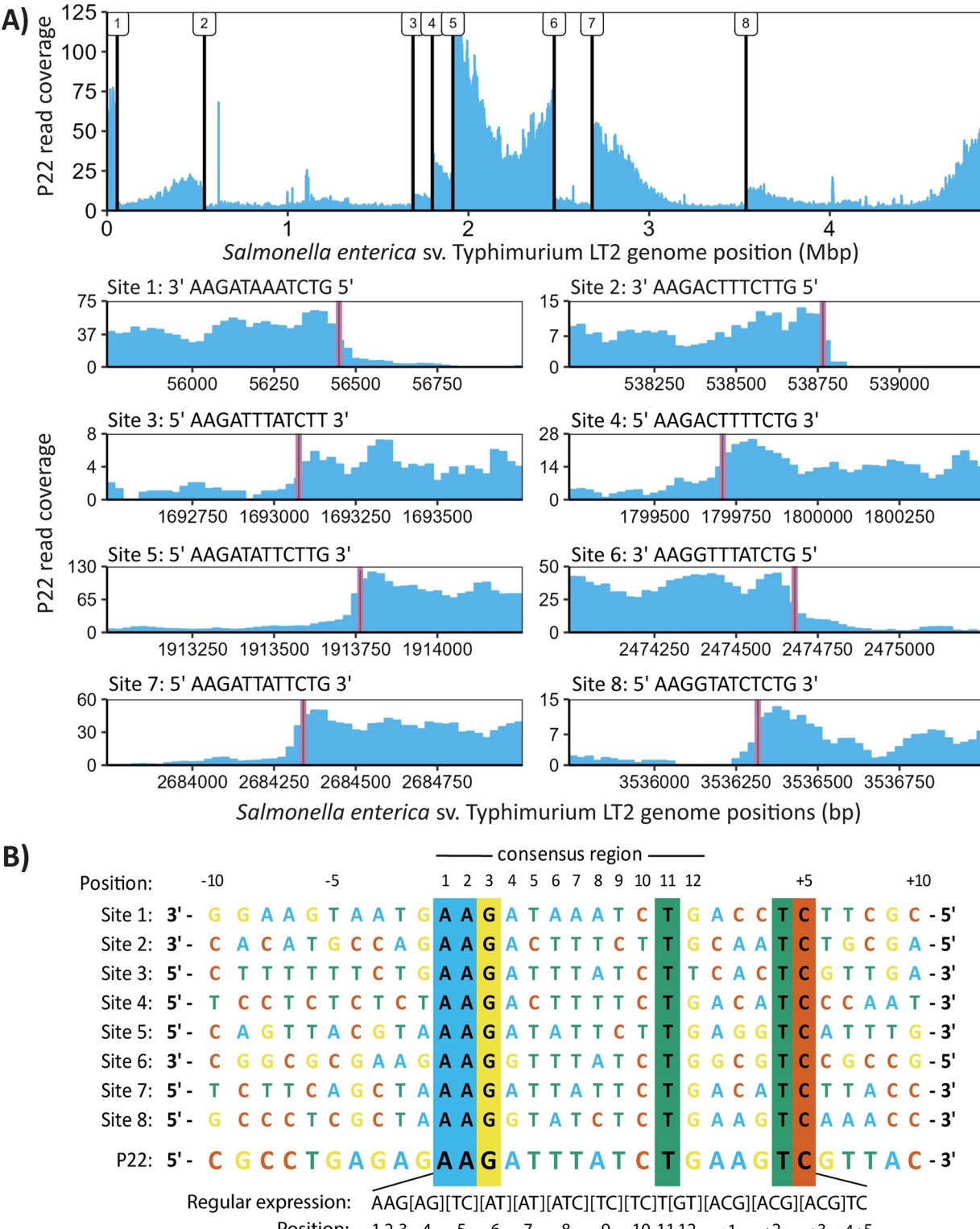

**Fig 1. *Salmonella* pseudo-pac site candidate sequences identified using P22 reads mapped to the *Salmonella* genome.** (A) Coverage plot of the *Salmonella enterica* Serovar Typhimurium LT2 (LT2) genome with sequencing reads from purified P22. Black vertical lines indicate the eight initiation sites for generalized transduction and the locations of the associated pseudo-pac sites along the LT2 genome. The exact locations of the pseudo-pac site sequences identified in this study are highlighted in pink on subsets of the LT2 read coverage plot associated with each site. The sequence of the pseudo-pac site candidate associated with each site is displayed above its respective read coverage plot. (B) A multiple sequence

alignment (MSA) generated with ClustalW of the 8 pseudo-pac site candidate sequences, the P22 pac site and the respective neighboring genome sequences. The location of the 12-bp pac site consensus region, identified in Casjens and colleagues [12] and further characterized by Wu and colleagues [13], is defined above the MSA. The regular expression pattern built from the pseudo-pac site candidate sequences is displayed below the MSA.

120 bp regions (Fig 1A) are henceforth referred to as pseudo-pac site candidates. We performed a multiple sequence alignment (MSA) of the pseudo-pac site candidates and neighboring genome regions which revealed that sequence conservation between the candidates extended beyond the 12 bp pseudo-pac site consensus region (Fig 1B). We adjusted the consensus region to include all strongly conserved regions of the MSA accordingly.

## Confirming accuracy of the pseudo-pac site consensus sequence

To determine if the consensus sequence obtained using the MSA was accurate, we used it to "scan" the genome of *Salmonella enterica* Serovar Typhimurium 14028 (14028S) for matches. Strain 14028S is susceptible to P22 infection and based on read mapping, we determined that 14028S shares the same generalized transduction sites and associated candidate pseudo-pac site sequences as LT2. However, when 14028S is infected with P22, we observed 1 additional generalized transduction site located on the reverse strand around 1.3 Mbp in the mapped read coverages. The additional site was seemingly not present in LT2 when infected with P22 (Fig 2A). We hypothesized that if our pseudo-pac site consensus sequence was correct, it could be used to identify the pseudo-pac site present at the additional generalized transduction site in 14028S. We built a regular expression (sequence match pattern)—5′ AAG[AG][TC][AT][AT][ATC][TC][TC]T[GT][ACG][ACG][ACG]TC 3′—that represents the bases observed for each position of the MSA. This regular expression specifies that any matching sequence will have an AAG in the first 3 positions, the fourth position must be either A or G, the fifth position must be either T or C, and so on. Five new sequences were identified in both LT2 and 14028S using the regular expression, two of which, located on the forward strand around 2.5 and 2.8 Mbp, were associated with right to left sloping read coverages immediately following the match location. These sequence matches likely represent 2 additional pseudo-pac sites in the LT2 and 14028S genomes (sites 9 and 10, respectively, in Fig 2C) which were not identified in our initial visual screen as they are not as prominent in LT2 compared to 14028S.

Despite the newly identified sites, the additional generalized transduction site present in 14028S was not identified by the regular expression which indicated an error in the consensus sequence. After testing various changes to the regular expression, we ultimately found that changing the conserved G in position three to G **or** C enabled the identification of the additional pseudo-pac site in 14028S (Fig 2C) with minimal false positives. While we only tested a relatively small number of all the possible changes to the regular expression, we found that changes to other conserved positions, like the As in the first 2 positions or the C in the last position, caused large increases in false positive matches (S1 Fig). Eight out of the 18 and 19 matches in LT2 and 14028S, respectively, to the final regular expression do not appear to be associated with large jumps and/or sloping read coverages. This could be due to the corresponding generalized transduction patterns being covered by more prominent patterns at these positions or by secondary DNA structures preventing the P22 packaging machinery from binding.

## Discussion

Based on the evidence presented, we are confident that the 10 pseudo-pac site sequences identified in LT2 (Fig 2D) are the exact sequences that P22 routinely recognizes for generalized

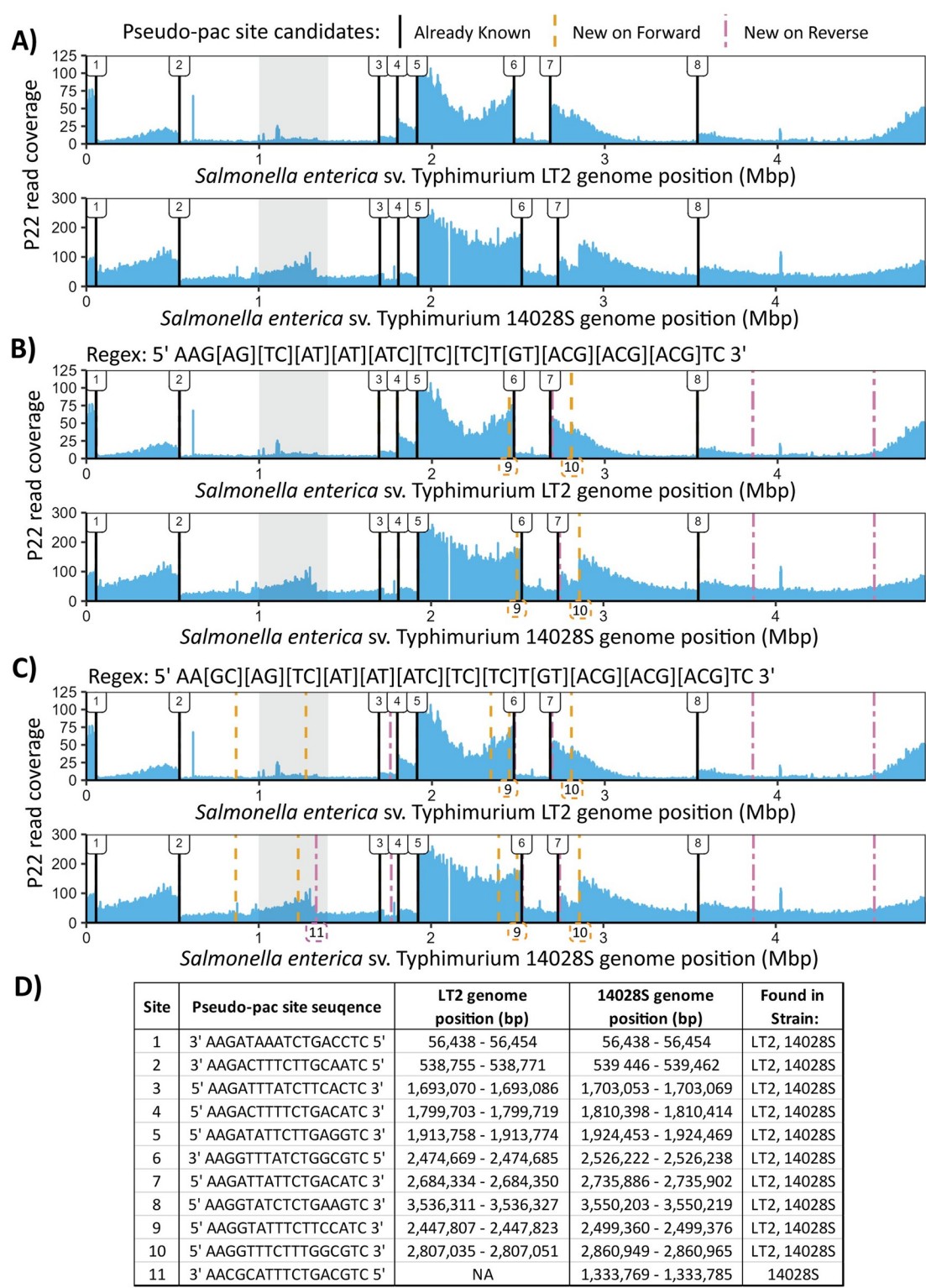

**Fig 2. *Salmonella* pseudo-pac site sequences identified using regular expression searches.** (A–C) Coverage plots of the *Salmonella enterica* Serovar Typhimurium LT2 (LT2) and *Salmonella enterica* Serovar Typhimurium 14028S (14028S) genomes with sequencing reads from purified P22. The additional generalized transduction site present in 14028S but not LT2 is shaded in grey. The regular expression (Regex) patterns used to search the *Salmonella* genomes for additional pseudo-pac sites are displayed above their associated plots. Black vertical lines indicate the locations of the pseudo-pac site sequences that were previously

identified in this study. Orange and pink dashed lines indicate the locations of regex matches on the forward and reverse Salmonella genome strands, respectively. The additional pseudo-pac sites identified with the regex searches, sites 9–11, are indicated in dashed boxes below their respective pattern match locations. (D) A summary table with information about each pseudo-pac site identified in this study numbered in order of detection. Sites 1–8 were initially visually identified using P22 read coverages mapped to the *Salmonella* genome. Sites 9–11 were identified using the regex searches and were confirmed visually with read coverages.

transduction. We are also confident that our final regular expression pseudo-pac site consensus sequence—5′ AA[GC][AG][TC][AT][AT][ATC][TC][TC]T[GT][ACG][ACG] [ACG]TC 3′—can identify highly efficient pseudo-pac sites in other P22-susceptible *Salmonella* strains, like the 11 sites identified in 14028S (Fig 2D). Our results could be further validated in vitro by genetically engineering the pseudo-pac site sequences identified in this study into a P22-susceptible host bacteria, infecting the host with P22, and sequencing the purified P22 to determine if generalized transduction was induced at the location of the inserted pseudo-pac site sequence. The identification of pseudo-pac sites in *Salmonella* provides fundamental insights into the sequence specificity necessary for P22 pac site recognition and opens the door to more targeted use of generalized transduction with P22.

Additionally, we hope that the methods used to identify the P22 pseudo-pac sites in LT2 and 14028S can be adapted by others to identify pseudo-pac sites used for generalized transduction in diverse phage—host systems.

## Materials and methods

The ultra-purified P22 reads used for mapping against LT2 originated from previously published Illumina sequencing reads [10]. We used BBMap [21] with ambiguous = random, qtrim = lr, and minid = 0.97 for mapping and pileup.sh with stdev = t and binsize = 25 to create data frames for read coverage visualization in R using ggplot2 [22]. We created an R script (S1 Text) to search the eight 120 bp regions across the LT2 genome for matches to the P22 pac site. We performed the MSA of both the P22 pac site and the pseudo-pac site candidates including the respective neighboring genome regions with ClustalW [23]. We used a Full Multiple Alignment and Bootstrap NJ Tree with 1,000 trees for the ClustalW MSA. We created an R function to search both the forward and reverse strands of genome sequences for regular expression matches (S1 Text).

## Supporting information

**S1 Fig. Examples of *Salmonella* genome matches to other regular expression patterns.** (A–C) Coverage plots of the *Salmonella enterica* sv. Typhimurium LT2 (LT2) and *Salmonella enterica* sv. Typhimurium 14028S (14028S) genomes with sequencing reads from purified P22. The additional generalized transduction site present in 14028S but not LT2 is shaded in gray. The regular expression (Regex) patterns used to search the *Salmonella* genomes for additional pseudo-pac sites are displayed above their associated plots. Black vertical lines indicate the locations of the pseudo-pac site sequences that were previously identified in this study. Orange and pink dashed lines indicate the locations of regular expression matches on the forward and reverse *Salmonella* genome strands, respectively.
(TIF)

**S1 Text. R code used to identify the pseudo-pac site sequences.**
(DOCX)

## Author Contributions

**Conceptualization:** Jessie L. Maier, Manuel Kleiner.

**Data curation:** Emma K. Sheriff, Breck A. Duerkop, Manuel Kleiner.

**Formal analysis:** Jessie L. Maier.

**Funding acquisition:** Breck A. Duerkop, Manuel Kleiner.

**Investigation:** Jessie L. Maier, Craig Gin, Benjamin Callahan, Emma K. Sheriff, Manuel Kleiner.

**Methodology:** Jessie L. Maier, Craig Gin, Benjamin Callahan, Manuel Kleiner.

**Project administration:** Manuel Kleiner.

**Resources:** Manuel Kleiner.

**Supervision:** Manuel Kleiner.

**Visualization:** Jessie L. Maier.

**Writing – original draft:** Jessie L. Maier.

**Writing – review & editing:** Jessie L. Maier, Craig Gin, Benjamin Callahan, Breck A. Duerkop, Manuel Kleiner.

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
