## [Decision Letter · Decision Letter 0]

3 May 2024

Dear Ms Maier,

Thank you very much for submitting your manuscript "Pseudo-pac site sequences used by phage P22 in generalized transduction of *Salmonella*" for consideration at PLOS Pathogens. Your manuscript was reviewed by members of the editorial board and by two independent reviewers. The reviewers appreciated the attention to an important topic. Based on the reviews, we are likely to accept this manuscript for publication, providing that you modify the manuscript according to the review recommendations.

Specifically, you need to address the issues noted with figure quality/resolution and sequence data availability for strain 14028S.

Sincerely,

Patrick Secor

Guest Editor

PLOS Pathogens

D. Scott Samuels

Section Editor

PLOS Pathogens

Michael Malim

Editor-in-Chief

PLOS Pathogens

orcid.org/0000-0002-7699-2064

Reviewer Comments (if any, and for reference):

Reviewer's Responses to Questions

**Part I - Summary**

Reviewer #1: The manuscript by Maier et al. is well written and answers an important question regarding what kind of host DNA sequences does P22 package. Based on analyzing the sequence reads from P22 genome sequencing data the authors were able to detect sequences that can act as pseudo-pac sites. Further, the predicted consensus sequence was refined by analyzing genome sequencing data of P22 on a different host. I have two minor issues that I would the authors to address to improve the presenation, rigor and reproducibility of the data.

Reviewer #2: This study focuses on the detection of pseudo-pac site sequences from Salmonella enterica Serovar Typhimurium. The team used existing sequencing data to identify where the pseudo-pac site facilitated LT2 genome packaging. These discovered sites were then used to identify those of another strain 14028S.

The method used to identify pseudo-pac sites is convincing, and several articles have already described the phenomenon of over-coverage. One way of improving would be to propose a method of defining consensus other than trial and error. However this would require a change of scale, with the analysis of data from a large number of phages and hosts.

Validation of the in vitro results proposed by the authors using a plasmid is a good way forward and I hope to see these results in a future article.

**Part II – Major Issues: Key Experiments Required for Acceptance**

Reviewer #1: None

Reviewer #2: No major modification

**Part III – Minor Issues: Editorial and Data Presentation Modifications**

Reviewer #1: The quality of the figures and the way they are presented. The figures are very pixelated and may be a better representation of the read abundance through out the genome will be make the impact of the paper more evident.

Is the illumina sequencing data for 14028S strain published ? If so please include a reference and if not please mention how it was done.

Reviewer #2: As a non-native English speaker, I can't comment on the English, but it seems perfectly comprehensible to me.

PLOS authors have the option to publish the peer review history of their article (what does this mean?). If published, this will include your full peer review and any attached files.

Reviewer #1: No

Reviewer #2: No

Figure Files:

Data Requirements:

Reproducibility:

References:

---

## [Editor Report · Decision Letter 1]

29 May 2024

Dear Ms Maier,

We are pleased to inform you that your manuscript 'Pseudo-pac site sequences used by phage P22 in generalized transduction of *Salmonella*' has been provisionally accepted for publication in PLOS Pathogens.

Best regards,

Patrick Secor

Guest Editor

PLOS Pathogens

D. Scott Samuels

Section Editor

PLOS Pathogens

Michael Malim

Editor-in-Chief

PLOS Pathogens

orcid.org/0000-0002-7699-2064
---

## [Editor Report · Acceptance letter]

13 Jun 2024

Dear Ms Maier,

We are delighted to inform you that your manuscript, "Pseudo-pac site sequences used by phage P22 in generalized transduction of *Salmonella*," has been formally accepted for publication in PLOS Pathogens.

Best regards,

Michael Malim

Editor-in-Chief

PLOS Pathogens

orcid.org/0000-0002-7699-2064